# Paternal country of origin and adverse neonatal outcomes in births to foreign-born women in Norway: A population-based cohort study

Eline S. Vik[1,2]*, Vigdis Aasheim[1], Roy M. Nilsen[1], Rhonda Small[3,4], Dag Moster[2,5], Erica Schytt[1,3,6]

1 Faculty of Health and Social Sciences, Western Norway University of Applied Sciences, Norway, 2 Department of Global Public Health and Primary Care, University of Bergen, Norway, 3 Department of Women's and Children's Health, Karolinska Institutet, Stockholm, Sweden, 4 Judith Lumley Centre, La Trobe University, Melbourne, Australia, 5 Department of Pediatrics, Haukeland University Hospital, Norway, 6 Centre for Clinical Research Dalarna, Uppsala University, Sweden

* eline.skirnisdottir.vik@hvl.no

**Data Availability Statement:** The data that support the findings of this study were obtained from the Medical Birth Registry of Norway and Statistics Norway and were used under license. Data may be

## Abstract

### Background

Migration is a risk factor for adverse neonatal outcomes. The various impacts of maternal origin have been reported previously. The aim of this study was to investigate associations between paternal origin and adverse neonatal outcomes in births to migrant and Norwegian-born women in Norway.

### Methods and findings

This nationwide population-based study included births to migrant (n = 240,759, mean age 29.6 years [±5.3 SD]) and Norwegian-born women (n = 1,232,327, mean age 29.0 years [±5.1 SD]) giving birth in Norway in 1990–2016. The main exposure was paternal origin (Norwegian-born, foreign-born, or unregistered). Neonatal outcomes were very preterm birth ($22^{+0}$–$31^{+6}$ gestational weeks), moderately preterm birth ($32^{+0}$–$36^{+6}$ gestational weeks), small for gestational age (SGA), low Apgar score (<7 at 5 minutes), and stillbirth. Associations were investigated in migrant and Norwegian-born women separately using multiple logistic regression and reported as adjusted odds ratios (aORs) with 95% confidence intervals (CIs), adjusted for year of birth, parity, maternal and paternal age, marital status, maternal education, and mother's gross income. In births to migrant women, a foreign-born father was associated with increased odds of very preterm birth (1.1% versus 0.9%, aOR 1.20; CI 1.08–1.33, p = 0.001), SGA (13.4% versus 9.5%, aOR 1.48; CI 1.43–1.53, p < 0.001), low Apgar score (1.7% versus 1.5%, aOR 1.14; CI 1.05–1.23, p = 0.001), and stillbirth (0.5% versus 0.3%, aOR 1.26; CI 1.08–1.48, p = 0.004) compared with a Norwegian-born father. In Norwegian-born women, a foreign-born father was associated with increased odds of SGA (9.3% versus 8.1%, aOR 1.13; CI 1.09–1.16, p < 0.001) and decreased odds of moderately preterm birth (4.3% versus 4.4%, aOR 0.95; CI 0.91–0.99, p

available from the authors upon reasonable request and with the permission of the Medical Birth Registry of Norway (datatilgang@fhi.no) and Statistics Norway (mikrodata@ssb.no).

**Funding:** Faculty of Health and Social Sciences (Western Norway University of Applied Sciences, Norway) was the main funder for this study. Centre for Clinical Research Dalarna (Uppsala University, Sweden) funded working hours for ES. Additional data costs were funded by the Norwegian SIDS and Stillbirth Society. None of the funders had a role in study design, data collection and analysis, decision to publish, or preparation of the manuscript.

**Competing interests:** The authors have declared that no competing interests exist.

**Abbreviations:** aOR, adjusted OR; CI, confidence interval; GBD, Global Burden of Disease; MBRN, Medical Birth Registry of Norway; OR, odds ratio; STROBE, Strengthening the Reporting of Observational Studies in Epidemiology.

= 0.015) when compared with a Norwegian-born father. In migrant women, unregistered paternal origin was associated with increased odds of very preterm birth (2.2% versus 0.9%, aOR 2.29; CI 1.97–2.66, p < 0.001), moderately preterm birth (5.6% versus 4.7%, aOR 1.15; CI 1.06–1.25, p = 0.001), SGA (13.0% versus 9.5%, aOR 1.50; CI 1.42–1.58, p < 0.001), low Apgar score (3.4% versus 1.5%, aOR 2.23; CI 1.99–2.50, p < 0.001), and stillbirth (1.5% versus 0.3%, aOR 4.87; CI 3.98–5.96, p < 0.001) compared with a Norwegian-born father. In Norwegian-born women, unregistered paternal origin was associated with increased odds of very preterm birth (4.6% versus 1.0%, aOR 4.39; CI 4.05–4.76, p < 0.001), moderately preterm birth (7.8% versus 4.4%, aOR 1.62; CI 1.53–1.71, p < 0.001), SGA (11.4% versus 8.1%, aOR 1.30; CI 1.24–1.36, p < 0.001), low Apgar score (4.6% versus 1.3%, aOR 3.51; CI 3.26–3.78, p < 0.001), and stillbirth (3.2% versus 0.4%, aOR 9.00; CI 8.15–9.93, p < 0.001) compared with births with a Norwegian-born father. The main limitations of this study were the restricted access to paternal demographics and inability to account for all lifestyle factors.

## Conclusion

We found that a foreign-born father was associated with adverse neonatal outcomes among births to migrant women, but to a lesser degree among births to nonmigrant women, when compared with a Norwegian-born father. Unregistered paternal origin was associated with higher odds of adverse neonatal outcomes in births to both migrant and nonmigrant women when compared with Norwegian-born fathers. Increased attention to paternal origin may help identify women in maternity care at risk for adverse neonatal outcomes.

## Author summary

### Why was this study done?

- International migration is increasing, and today, 1 in 7 of the world's population is a migrant.

- It is well documented that subgroups of migrant women are at increased risk of adverse neonatal outcomes.

- Few studies have examined the association between paternal origin on adverse neonatal outcomes in migrant women.

### What did the researchers do and find?

- We investigated associations between paternal origin and adverse neonatal outcomes in births to migrant and Norwegian-born women who gave birth in Norway between 1990 and 2016.

- Separate analyses were conducted for 240,759 births to migrant women and, for comparison, 1,232,327 births to Norwegian-born women.

- In births to migrant women, a foreign-born or unregistered father of the child was associated with higher odds of adverse outcomes when compared with a Norwegian-born father.

- In births to Norwegian-born women, a foreign-born father of the child was associated with increased odds of small for gestational age (SGA) and decreased odds of moderately preterm birth when compared to births for which the father was Norwegian-born.

### What do these findings mean?

- Foreign paternal origin is associated with adverse neonatal outcomes in births to migrant women.

- Greater focus on missing information on paternal origin may help identify women in maternity care at risk for adverse neonatal outcomes.

## Introduction

With an increasing number of babies born to migrant parents, the needs of migrant families in maternity care have been declared a priority for research and action by the World Health Organization [1]. Migrant women have been identified with an increased risk of adverse neonatal outcomes, such as low birthweight [2], small for gestational age (SGA) [3,4], preterm birth [2,5], and perinatal morbidity and mortality [2,6,7]. However, most studies focus on the impact of maternal factors, and less attention has been paid to paternal factors.

Both parents' origins seem to influence pregnancy outcomes [3,6,8–11]. However, the number of studies investigating paternal origin is limited, and the methodological differences between studies and the complexity of migration all make the findings difficult to interpret [3,6,8–11]. Large population-based studies from Norway, Sweden, Canada, and Australia have shown that a partner from the host population is associated with a decreased risk of very preterm birth [9,10], SGA [3], and stillbirth [6,8,11]. However, the increased risks of preterm birth in general [3,10], moderately preterm birth [9,10], and low Apgar score [3] have been observed irrespective of whether the father is from the host population or not. Further, unregistered paternal origin has been associated with an increased risk of stillbirth [6], and lack of paternal data is acknowledged as an important factor for identifying high-risk pregnant women [12–14]. Other paternal factors associated with neonatal outcomes have also been reported, such as advanced paternal age seeming to increase the risk of preterm birth [15] and stillbirth [16]. Understanding the impact of paternal factors on adverse birth outcomes is not straightforward. There is a need for population-based studies investigating associations between paternal factors, including births when paternal data are missing, and a wider range of adverse neonatal outcomes from a migration perspective.

In this nationwide study, we had access to a standardized collection of population-based data on both maternal and paternal factors, information that is often limited in other studies. To contribute to the understanding of the impact of paternal origin, we investigated the associations between paternal origin (Norwegian-born, foreign-born, or unregistered) and adverse neonatal outcomes (very preterm birth, moderately preterm birth, low Apgar score, and stillbirth) in births to migrant and Norwegian-born women giving birth in Norway between 1990 and 2016. Our hypothesis was that a foreign-born father of the child would be associated with an increased risk of adverse pregnancy outcomes in births to migrant women.

## Methods

### Study design

This population-based register study included births to migrant women giving birth in Norway between the 1st of January 1990 and 31st of December 2016. Data from the Medical Birth Registry of Norway (MBRN) [17] were linked with data provided by Statistics Norway [18] using each woman's unique national identity number. Statistics Norway provided data on paternal and maternal country of birth, as well as maternal socioeconomic-related factors. The MBRN provided information on paternal identity and paternal age, as well as data on maternal and infant health, including detailed information on previous pregnancies and births in Norway, though not on births outside of Norway. There was no prospective protocol or analysis plan. The reporting of this study follows the Strengthening the Reporting of Observational Studies in Epidemiology (STROBE) guideline (S1 STROBE Checklist) [19].

### Study population

This study included births to migrant women (n = 240,759 births) and Norwegian-born women (n = 1,232,327) giving birth in Norway in the period 1990–2016 (Fig 1). Migrant women were defined as foreign-born women with 2 foreign-born parents. Births to women born to migrant parents and women with mixed background (migrant women registered with 1 or 2 Norwegian-born parents) or with unknown background, in total 5% of the birth cohort, were excluded to reduce heterogeneity across comparison groups and to simplify interpretation of the results.

### Setting

In 2018, 29% of all births in Norway were to mothers with a migrant background [20]. Maternity care in Norway is based on the individual woman's medical needs, and care during

**Included births**

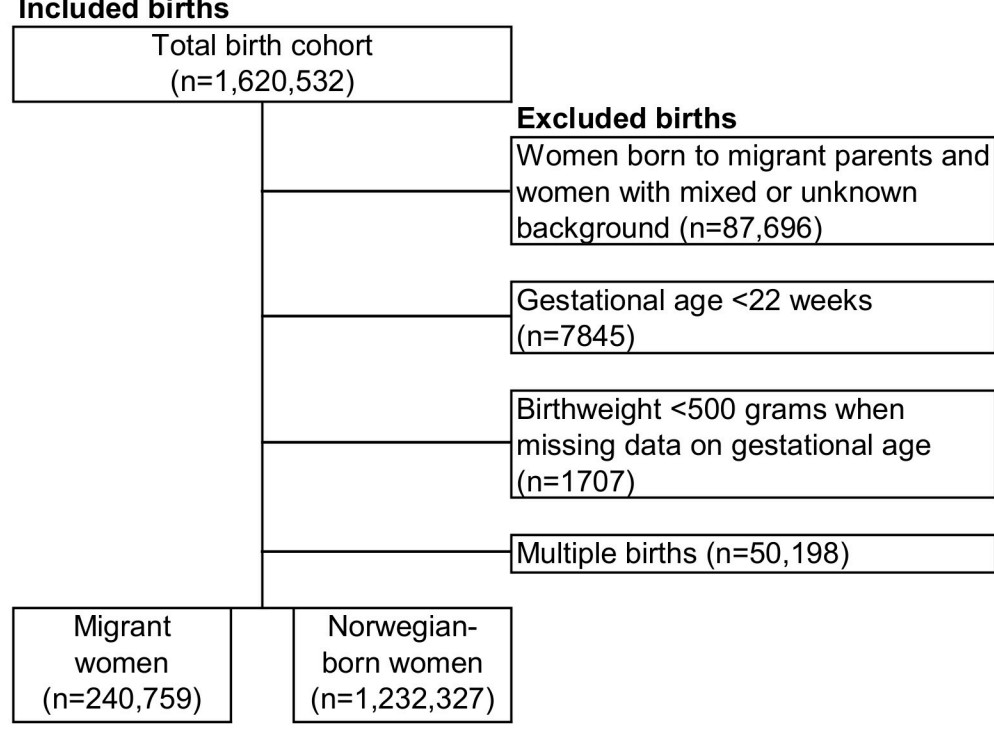

**Fig 1. Flowchart of the derivation of the study sample (n = 1,620,532).**

pregnancy and birth is free of charge [21]. The healthcare system is considered of high quality, with low maternal and child mortality ratios [22]. However, there are inequalities in healthcare, and migrant women in Scandinavian countries have been found to receive suboptimal care compared with nonmigrant women [23,24]. In Norway, migrant families often struggle economically, and children living in poverty are often children of migrant parents [25].

## Paternal factors

Paternal origin was based on information on paternal country of birth retrieved from Statistics Norway (Norwegian-born, foreign-born, or unregistered). Explanation for not being registered in the Statistics Norway may be that the baby's father was unknown to the mother or that the baby's father was not a citizen of Norway but still known to the mother. In an effort to distinguish between known and unknown paternal identity, we analyzed a novel variable as derived by the MBRN. In the MBRN, paternal identity (known, unknown) is registered as known when the father's national identity number or his date of birth has been given by the mother or retrieved from Statistics Norway via routine updates; otherwise, his identity is registered as unknown. By collecting both maternal report and data from Statistics Norway, categories of paternal identity may overlap largely with paternal origin (Norwegian-born, foreign-born, or unregistered), but they will still be different in content and therefore analyzed separately. The MBRN also included data on paternal age.

## Paternal and maternal region of birth

Paternal and maternal country of birth were categorized into the following regions according to the classification of Global Burden of Disease (GBD) [26]: high-income countries; Central and Eastern Europe, Central Asia; North Africa and the Middle East; sub-Saharan Africa; South Asia; Southeast Asia, East Asia, and Oceania; and Latin America and the Caribbean.

## Maternal factors

The MBRN provided data on maternal age, marital status (single, married/cohabiting, divorced/separated/widowed, other/not given), and parity (0, 1, 2, 3, or ≥4 previous births). Statistics Norway provided data on maternal level of education (no education, primary school, secondary school, university/college, missing) and mother's gross income (categorized into quartiles, missing).

## Adverse neonatal outcomes

The main outcomes were very preterm birth, moderately preterm birth, SGA, low Apgar score, and stillbirth. Gestational age was based on ultrasound estimation or, when such information was lacking, calculated from the first day of the last menstrual period. In the analyses of very preterm birth ($22^{+0}$–$31^{+6}$ gestational weeks) and moderately preterm birth ($32^{+0}$–$36^{+6}$ gestational weeks), term birth, including post-term birth ($\geq37^{+0}$ gestational weeks), was used as the comparison group. In analyses of very preterm birth, moderately preterm births were excluded, and in analysis of moderately preterm birth, very preterm births were excluded. For calculating SGA, we used Norwegian standards combining information on gestational age, birthweight, and sex of the child [27]. In the analyses of SGA, we excluded births with missing data on any of those 3 variables. Low Apgar score was defined as <7 at 5 minutes, which is commonly used and regarded as clinically relevant [28,29]. Stillbirth was defined as a pregnancy loss at ≥22 weeks of gestation (or with a birthweight ≥500 g if data on gestational age were missing).

## Statistical methods

To examine the associations between paternal origin and neonatal outcomes, we estimated odds ratios (ORs) with 95% confidence intervals (CIs) using binary logistic regression models. Adjustments were made for year of birth, parity, maternal age, paternal age, marital status, maternal education, and mother's gross income. To account for nonlinear relations, year of birth, maternal age, and paternal age were incorporated as polynomial quadratic model terms. To account for dependency between births by the same mother, we used robust standard errors that allowed for within-mother clustering.

To avoid listwise deletion and potential bias due to missing data in the covariates education and income, we performed multiple imputations on missing data, assuming that the missing data could be explained by other variables in the data set (missing at random). The imputation algorithm used was multivariate normal [30], and a total of 5 imputed data sets were created. Separate imputation models were used for each outcome and included the 4 respective outcomes, maternal country of birth, and exposure variables (paternal origin and paternal identity), as well as all adjustment variables included in the final analytical models.

An a priori strategy of analyzing data separately for primiparous and multiparous women was initially chosen, based on the fact that primiparous and multiparous women are often provided with different recommendations in antenatal guidelines [31]. Therefore, we believed that this difference could affect associations differently for these groups. Because there was no strong indication that the associations did differ between these groups in relation to paternal origin, we reanalyzed the data and report the associations for primiparous and multiparous women combined.

All analyses were performed using Stata IC version 16 (Stata Statistical Software, College Station, TX, USA) for Windows.

## Ethics

The Regional Committees for Medical and Health Research Ethics approved this study, reference number: 2014/1278/REK South-East, Norway. A pseudonymous identity number was generated for each individual in the data set prior to access by the authors.

## Results

The final sample included births to migrant (n = 240,759) and Norwegian-born (n = 1,232,327) women giving birth in Norway between the years 1990 and 2016. The women and men originated from 202 and 203 different countries, respectively, including Norway. In births to migrant women, the 10 most common paternal countries of birth were Norway (28.9%), Somalia (5.4%), Pakistan (5.1%), Iraq (4.6%), Poland (4.5%), Vietnam (3.0%), Sri Lanka (2.7%), Turkey (2.5%), Kosovo (2.4%), and Lithuania (2.0%). Similar, the 10 most common maternal countries of birth were Somalia (6.8%), Poland (6.4%), Sweden (6.3%), Pakistan (6.1%), Iraq (4.6%), the Philippines (4.0%), Vietnam (3.7%), Thailand (3.3%), Russia (2.9%), and Turkey (2.8%). In births to Norwegian-born women, the 10 most common paternal countries of birth were Norway (92.2%), Sweden (1.0%), United Kingdom (0.6%), Denmark (0.5%), the United States of America (0.5%), Germany (0.3%), Turkey (0.2%), the Netherlands (0.2%), Chile (0.1%), and Iran (0.1%).

Table 1 shows the background characteristics of the sample of births to migrant and Norwegian-born women by paternal origin (Norwegian-born, foreign-born, and unregistered). In births to migrant women and Norwegian-born fathers (29%), the women most commonly originated from high-income countries or from Southeast Asia, East Asia, or Oceania; a majority of the mothers were married or cohabiting, had a high level of education, and belonged to

**Table 1. Background characteristics by paternal origin in births to migrant (n = 240,759) and Norwegian-born (n = 1,232,327) women in Norway (1990–2016).**

| | Migrant Women | | | | Norwegian-Born Women | | | |
|---|---|---|---|---|---|---|---|---|
| | Paternal Origin | | | | | | | |
| | Norwegian-born | Foreign-born | Unregistered | | Norwegian-born | Foreign-born | Unregistered | |
| | n (%) | n (%) | n (%) | P-value | n (%) | n (%) | n (%) | P-value |
| Total | 69,477 (28.9) | 152,097 (63.2) | 19,185 (8.0) | | 1,135,712 (92.2) | 73,763 (6.0) | 22,852 (1.9) | |
| **Paternal factors** | | | | | | | | |
| Paternal age, years (mean ± SD) | 35.7 ± 7.6 | 33.7 ± 6.5 | 36.4 ± 8.2 | | 31.6 ± 5.8 | 32.7 ± 6.2 | 34.0 ± 6.6 | |
| Paternal age, missing | 91 (0.1) | 429 (0.3) | 5,615 (29.3) | | 1,293 (0.1) | 234 (0.3) | 7,498 (32.9) | |
| Paternal origin (GBD) | | | | <0.001 | | | | <0.001 |
| *High-income countries* | 69,477 (100.0) | 19,454 (12.8) | | | 1,135,712 (100.0) | 50,423 (68.4) | | |
| *Central and Eastern Europe, Central Asia* | | 36,648 (24.1) | | | | 4,270 (5.8) | | |
| *North Africa, Middle East* | | 35,594 (23.4) | | | | 7,353 (10.0) | | |
| *Sub-Saharan Africa* | | 24,064 (15.8) | | | | 4,599 (6.2) | | |
| *Southeast Asia, East Asia, Oceania* | | 19,053 (12.5) | | | | 2,383 (3.2) | | |
| *South Asia* | | 16,175 (10.6) | | | | 1,449 (2.0) | | |
| *Latin America, the Caribbean* | | 1,109 (0.7) | | | | 3,274 (4.4) | | |
| Paternal identity unknown‡ | 70 (0.1) | 220 (0.1) | 1,798 (9.4) | <0.001 | 1,171 (0.1) | 131 (0.2) | 4,857 (21.3) | <0.001 |
| **Maternal factors** | | | | | | | | |
| Maternal age, years (mean ± SD) | 30.5 ± 5.0 | 29.2 ± 5.2 | 30.3 ± 5.9 | | 28.9 ± 5.1 | 30.2 ± 5.3 | 30.4 ± 6.3 | |
| Marital status | | | | <0.001 | | | | <0.001 |
| *Single* | 2,035 (2.9) | 6,163 (4.1) | 4,452 (23.2) | | 67,517 (5.9) | 7,016 (9.5) | 8,486 (37.1) | |
| *Married* | 47,842 (68.9) | 124,274 (81.7) | 10,522 (54.8) | | 505,758 (44.5) | 36,758 (49.8) | 6,222 (27.2) | |
| *Cohabiting* | 18,422 (26.5) | 17,523 (11.5) | 2,919 (15.2) | | 549,542 (48.4) | 28,548 (38.7) | 7,402 (32.4) | |
| *Divorced/separated/widowed* | 510 (0.7) | 1,958 (1.3) | 786 (4.1) | | 4,317 (0.4) | 738 (1.0) | 400 (1.8) | |
| *Other/not given* | 668 (1.0) | 2,179 (1.4) | 506 (2.6) | | 8,578 (0.8) | 703 (1.0) | 342 (1.5) | |
| Maternal origin (GBD) | | | | <0.001 | | | | <0.001 |
| *High-income countries* | 31,350 (45.1) | 17,890 (11.8) | 2,282 (11.9) | | 1,135,712 (100.0) | 73,763 (100.0) | 22,852 (100.0) | |
| *Central and Eastern Europe, Central Asia* | 12,120 (17.4) | 39,066 (25.7) | 4,113 (21.4) | | | | | |
| *North Africa, Middle East* | 2,094 (3.0) | 34,108 (22.4) | 2,001 (10.4) | | | | | |
| *Sub-Saharan Africa* | 2,317 (3.3) | 23,544 (15.5) | 5,930 (30.9) | | | | | |
| *Southeast Asia, East Asia, Oceania* | 14,931 (21.5) | 20,094 (13.2) | 3,537 (18.4) | | | | | |
| *South Asia* | 2,495 (3.6) | 16,052 (10.6) | 533 (2.8) | | | | | |
| *Latin America, the Caribbean* | 4,170 (6.0) | 1,343 (0.9) | 789 (4.1) | | | | | |
| Maternal education§ | | | | <0.001 | | | | <0.001 |
| *No education* | 420 (0.7) | 3,908 (3.6) | 774 (5.7) | | 36 (0.0) | 3 (0.0) | 3 (0.0) | |
| *Primary education* | 10,615 (18.6) | 37,872 (35.2) | 6,205 (46.0) | | 236,069 (20.8) | 13,832 (18.8) | 7,375 (32.6) | |
| *Secondary school* | 14,509 (25.4) | 28,650 (26.6) | 3,231 (24.0) | | 429,431 (37.9) | 22,266 (30.3) | 7,669 (33.9) | |
| *University/college* | 31,603 (55.3) | 37,256 (34.6) | 3,283 (24.3) | | 468,317 (41.3) | 37,423 (50.9) | 7,582 (33.5) | |
| Maternal education, missing | 12,330 (17.8) | 44,411 (29.2) | 5,692 (29.7) | | 1,859 (0.2) | 239 (0.3) | 223 (1.0) | |
| Mother's gross income§ | | | | <0.001 | | | | <0.001 |
| *≤25th percentile* | 14,221 (25.8) | 34,053 (35.7) | 5,271 (44.5) | | 255,270 (24.3) | 13,218 (19.7) | 5,195 (27.0) | |
| *25th–50th percentile* | 10,398 (18.9) | 17,524 (18.4) | 2,042 (17.2) | | 276,078 (26.2) | 13,842 (20.6) | 3,942 (20.5) | |
| *50th–75th percentile* | 13,541 (24.6) | 21,953 (23.0) | 2,417 (20.4) | | 266,901 (25.4) | 17,202 (25.6) | 4,321 (22.5) | |
| *≥75th percentile* | 16,886 (30.7) | 21,857 (22.9) | 2,114 (17.9) | | 254,270 (24.2) | 22,951 (34.2) | 5,790 (30.1) | |
| Mother's gross income, missing | 14,431 (20.8) | 56,710 (37.3) | 7,341 (38.3) | | 83,193 (7.3) | 6,550 (8.9) | 3,604 (15.8) | |
| Parity | | | | <0.001 | | | | <0.001 |
| *0* | 34,684 (49.9) | 59,294 (39.0) | 4,919 (25.6) | | 467,831 (41.2) | 33,359 (45.2) | 8,126 (35.6) | |
| *1* | 23,784 (34.2) | 48,826 (32.1) | 7,522 (39.2) | | 416,257 (36.7) | 26,086 (35.4) | 5,169 (22.6) | |

(*Continued*)

**Table 1.** (Continued)

| | Migrant Women | | | | Norwegian-Born Women | | | |
|---|---|---|---|---|---|---|---|---|
| | Paternal Origin | | | | | | | |
| | Norwegian-born | Foreign-born | Unregistered | | Norwegian-born | Foreign-born | Unregistered | |
| | n (%) | n (%) | n (%) | P-value | n (%) | n (%) | n (%) | P-value |
| 2 | 8,245 (11.9) | 24,440 (16.1) | 3,604 (18.8) | | 188,511 (16.6) | 10,490 (14.2) | 5,228 (22.9) | |
| 3 | 1,983 (2.9) | 10,611 (7.0) | 1,814 (9.5) | | 46,399 (4.1) | 2,708 (3.7) | 3,020 (13.2) | |
| ≥4 | 781 (1.1) | 8,926 (5.9) | 1,326 (6.9) | | 16,714 (1.5) | 1,120 (1.5) | 1,309 (5.7) | |

**Abbreviations:** GBD, Global Burden of Disease; SD, standard deviation.

‡A known father may be either foreign-born or Norwegian-born.

§Percentages are calculated from nonmissing data.

the highest maternal gross income categories, and almost half of the women were primiparous. In births to migrant women and foreign-born fathers (63%), both parents mostly originated from Central and Eastern Europe and Central Asia or North Africa and the Middle East; a majority of women were married or cohabiting and had low levels of education (or missing information) and maternal gross income, and 39% were primiparous. When paternal origin was unregistered (8%), paternal age and paternal identity was also missing for 29% and 9% of the births, respectively; the mothers commonly originated from sub-Saharan Africa or Central and Eastern Europe and Central Asia, many were single and had low levels of education (or missing information) and maternal gross income, and 26% were primiparous.

In births to Norwegian-born women and Norwegian-born fathers (92%), the mothers often had high levels of education and maternal gross income. In births to Norwegian-born women and foreign-born fathers (6%), the fathers commonly originated from high-income countries and North Africa and the Middle East, and the mothers had high levels of education and maternal gross income. If paternal origin was unregistered (2%), paternal age was often missing, paternal identity was often unknown, and the mothers often had low levels of education, had missing data on maternal gross income, and had high parity.

Fig 2 shows paternal origin (Norwegian-born, foreign-born, or unregistered) in relation to maternal region of birth. In births to migrant women, a Norwegian-born father was common among women from high-income countries; Southeast Asia, East Asia, and Oceania; and Latin America and the Caribbean, and a foreign-born father was common among women from North Africa and the Middle East and South Asia as well as in women from sub-Saharan Africa. Unregistered paternal origin was common in births to migrant women from sub-Saharan Africa (19%); Latin America and the Caribbean (13%); Southeast Asia, East Asia, and Oceania (9%); and Central and Eastern Europe and Central Asia (7%). In births to Norwegian-born women, paternal origin was registered as Norwegian-born and foreign-born in 92% and 6%, respectively. Paternal origin was unregistered in 2% of births to Norwegian-born women.

Table 2 shows the associations between paternal origin and adverse neonatal outcomes in births to migrant women. Overall, the results only changed slightly after adjustments for year of birth, parity, maternal age, paternal age, marital status, maternal education, and mother's gross income. In births to migrant women, a foreign-born father was associated with increased odds of very preterm birth (adjusted OR [aOR] 1.20; CI 1.08–1.33), SGA (aOR 1.48; CI 1.43–1.53), low Apgar score (aOR 1.14; CI 1.05–1.23), and stillbirth (aOR 1.26; CI 1.08–1.48) compared with a Norwegian-born father. In births for which paternal origin was unregistered, the odds were increased for very preterm birth (aOR 2.29; CI 1.97–2.66), moderately preterm

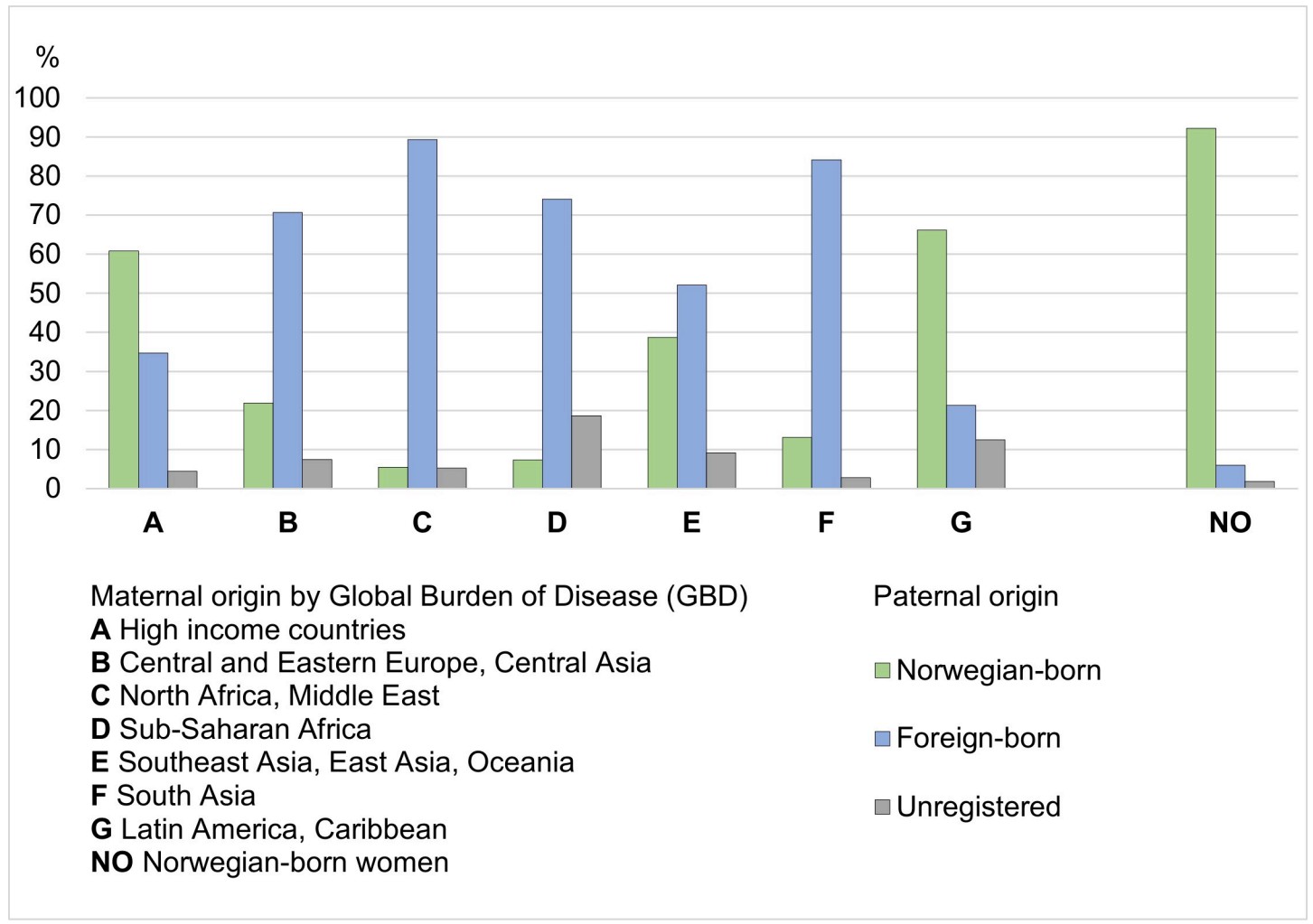

**Fig 2. Proportion of births to migrant and Norwegian-born women giving birth in Norway (1990–2016) by maternal region of birth (GBD categories; Norwegian-born women in a separate category) and paternal origin (Norwegian-born, foreign-born, and unregistered).** GBD, Global Burden of Disease.

birth (aOR 1.15; CI 1.06–1.25), SGA (aOR 1.50; CI 1.42–1.58), low Apgar score (aOR 2.23; CI 1.99–2.50), and stillbirth (aOR 4.87; CI 3.98–5.96) compared with births with a Norwegian-born father.

Table 3 shows the associations between paternal origin and adverse neonatal outcomes in births to Norwegian-born women. In births to Norwegian-born women, a foreign-born father was associated with increased odds of SGA (aOR 1.13; CI 1.09–1.16) and decreased odds of moderately preterm birth (aOR 0.95; CI 0.91–0.99) compared with a Norwegian-born father. In births for which paternal origin was unregistered, the odds were increased for very preterm birth (aOR 4.39; CI 4.05–4.76), moderately preterm birth (aOR 1.62; CI 1.53–1.71), SGA (aOR 1.30; CI 1.24–1.36), low Apgar score (aOR 3.51; CI 3.26–3.78), and stillbirth (aOR 9.00; CI 8.15–9.93) compared with births with a Norwegian-born father.

In births to migrant women, unknown paternal identity was associated with increased odds of very preterm birth (aOR 10.36; CI 8.04–13.36), moderately preterm birth (aOR 1.26; CI 1.03–1.54), SGA (aOR 1.35; CI 1.18–1.54), low Apgar score (aOR 3.26; CI 2.61–4.06), and stillbirth (aOR 16.62; CI 13.62–20.28) compared with known paternal identity (S1 Table). In births

**Table 2. Associations between paternal origin and adverse neonatal outcomes in births to migrant women in Norway (1990–2016).**

| | Very Preterm (22$^{+0}$–31$^{+6}$ gwks)† | Moderately Preterm (32$^{+0}$–36$^{+6}$ gwks)‡ | SGA§ | Apgar Score <7 at 5 Minutes\$ | Stillbirth |
|---|---|---|---|---|---|
| **Paternal origin** | | | | | |
| Norwegian-born (n) | 64,818 | 67,369 | 67,925 | 69,246 | 69,477 |
| No cases (%) | 615 (0.9) | 3,166 (4.7) | 6,466 (9.5) | 1,043 (1.5) | 229 (0.3) |
| Reference | 1.00 | 1.00 | 1.00 | 1.00 | 1.00 |
| Foreign-born (n) | 141,697 | 146,911 | 148,350 | 151,538 | 152,097 |
| No cases (%) | 1,573 (1.1) | 6,787 (4.6) | 19,836 (13.4) | 2,606 (1.7) | 695 (0.5) |
| OR (95% CI) | 1.17 (1.06–1.29) | 0.98 (0.94–1.03) | 1.47 (1.42–1.52) | 1.14 (1.06–1.23) | 1.39 (1.19–1.61) |
| P-value | 0.001 | 0.443 | <0.001 | <0.001 | <0.001 |
| aOR (95% CI)¶ | 1.20 (1.08–1.33) | 0.98 (0.93–1.03) | 1.48 (1.43–1.53) | 1.14 (1.05–1.23) | 1.26 (1.08–1.48) |
| P-value | 0.001 | 0.330 | <0.001 | 0.001 | 0.004 |
| Unregistered (n) | 17,656 | 18,285 | 18,634 | 19,062 | 19,185 |
| No cases (%) | 386 (2.2) | 1,015 (5.6) | 2,420 (13.0) | 644 (3.4) | 294 (1.5) |
| OR (95% CI) | 2.33 (2.05–2.66) | 1.19 (1.11–1.28) | 1.42 (1.35–1.49) | 2.29 (2.07–2.53) | 4.71 (3.96–5.59) |
| P-value | <0.001 | <0.001 | <0.001 | <0.001 | <0.001 |
| aOR (95% CI)¶ | 2.29 (1.97–2.66) | 1.15 (1.06–1.25) | 1.50 (1.42–1.58) | 2.23 (1.99–2.50) | 4.87 (3.98–5.96) |
| P-value | <0.001 | 0.001 | <0.001 | <0.001 | <0.001 |

**Abbreviations:** aOR, adjusted OR; CI, confidence interval; gwk, gestational week; OR, odds ratio; SGA, small for gestational age.

†Cases with missing data on gestational age (n = 5,620) and moderately preterm births (n = 10,968) excluded.

‡Cases with missing data on gestational age (n = 5,620) and very preterm births (n = 2,574) excluded.

§Cases with missing data on SGA excluded (n = 5,850).

\$Cases with missing data on Apgar score excluded (n = 913).

¶Adjusted for year of birth, parity, maternal age, paternal age, marital status, maternal education, and mother's gross income.

to Norwegian-born women, unknown paternal identity was associated with increased odds of very preterm birth (aOR 15.43; CI 13.63–17.46), moderately preterm birth (aOR 1.58; CI 1.42–1.75), SGA (aOR 1.49; CI 1.38–1.60), low Apgar score (aOR 5.77; CI 5.09–6.55), and stillbirth (aOR 28.52; CI 25.83–31.49) compared with known paternal identity (S1 Table).

## Discussion

Our hypothesis that a foreign-born father of the child would be associated with an increased risk of adverse pregnancy outcomes in births to migrant women was confirmed. In births to migrant women, a foreign-born father was associated with increased odds of very preterm birth, SGA, low Apgar score, and stillbirth compared with a Norwegian-born father. Unregistered paternal country of origin was associated with increased odds of all adverse outcomes investigated in migrant women (very preterm birth, moderately preterm birth, SGA, low Apgar score, and stillbirth) compared with a Norwegian-born father. In births to Norwegian-born women, a foreign-born father was associated with increased odds of SGA and decreased odds of moderately preterm birth when compared with a Norwegian-born father. However, unregistered paternal origin also increased the odds of all adverse outcomes in this group of women.

The main strengths of this study include the standardized collection of paternal, maternal, and infant data [17,18] in a population including nearly all births to migrant women in Norway over a period of 26 years. Limitations include lack of data on previous preterm births, which is a known predictor for recurrent preterm delivery [32]. Furthermore, although lifestyle factors may be implicated in adverse outcomes, access to such information was limited in

**Table 3. Associations between paternal origin and adverse neonatal outcomes in births to Norwegian-born women in Norway (1990–2016).**

| | Very Preterm (22$^{+0}$–31$^{+6}$ gwks)† | Moderately Preterm (32$^{+0}$–36$^{+6}$ gwks)‡ | SGA§ | Apgar Score <7 at 5 Minutes$ | Stillbirth |
|---|---|---|---|---|---|
| **Paternal origin** | | | | | |
| Norwegian-born (n) | 1,045,115 | 1,083,099 | 1,092,170 | 1,129,567 | 1,135,712 |
| No cases (%) | 10,025 (1.0) | 48,009 (4.4) | 88,693 (8.1) | 14,714 (1.3) | 4,356 (0.4) |
| Reference | 1.00 | 1.00 | 1.00 | 1.00 | 1.00 |
| Foreign-born (n) | 68,475 | 70,854 | 71,470 | 73,416 | 73,763 |
| No cases (%) | 674 (1.0) | 3,053 (4.3) | 6,628 (9.3) | 995 (1.4) | 281 (0.4) |
| OR (95% CI) | 1.03 (0.95–1.11) | 0.97 (0.93–1.01) | 1.16 (1.12–1.19) | 1.04 (0.98–1.11) | 0.99 (0.88–1.12) |
| P-value | 0.527 | 0.140 | <0.001 | 0.226 | 0.912 |
| aOR (95% CI)¶ | 1.03 (0.95–1.12) | 0.95 (0.91–0.99) | 1.13 (1.09–1.16) | 0.98 (0.91–1.04) | 1.03 (0.91–1.16) |
| P-value | 0.504 | 0.015 | <0.001 | 0.466 | 0.687 |
| Unregistered (n) | 20,180 | 20,883 | 21,724 | 22,485 | 22,852 |
| No cases (%) | 926 (4.6) | 1,629 (7.8) | 2,483 (11.4) | 1,034 (4.6) | 739 (3.2) |
| OR (95% CI) | 4.97 (4.63–5.32) | 1.82 (1.73–1.92) | 1.46 (1.40–1.52) | 3.65 (3.42–3.90) | 8.68 (8.00–9.41) |
| P-value | <0.001 | <0.001 | <0.001 | <0.001 | <0.001 |
| aOR (95% CI)¶ | 4.39 (4.05–4.76) | 1.62 (1.53–1.71) | 1.30 (1.24–1.36) | 3.51 (3.26–3.78) | 9.00 (8.15–9.93) |
| P-value | <0.001 | <0.001 | <0.001 | <0.001 | <0.001 |

**Abbreviations:** aOR, adjusted OR; CI, confidence interval; gwk, gestational week; OR, odds ratio; SGA, small for gestational age.

†Cases with missing data on gestational age (n = 45,866) and moderately preterm births (n = 52,691) excluded.

‡Cases with missing data on gestational age (n = 45,866) and very preterm births (n = 11,625) excluded.

§Cases with missing data on SGA excluded (n = 46,963).

$Cases with missing data on Apgar score excluded (n = 6,859).

¶Adjusted for year of birth, parity, maternal age, paternal age, marital status, maternal education, and mother's gross income.

the current study. On the other hand, because many migrant women in high-income countries appear to have some healthier lifestyle habits than women born in these countries, such as less use of alcohol and tobacco [33,34], adverse outcomes for migrant women may be less likely to be attributable to these factors. Unfortunately, we did not have access to information on paternal education or income. Additional information on paternal education may have explained some of the increased risk because such information is acknowledged to be an independent risk factor for certain adverse neonatal outcomes such as preterm birth [35,36]. To better understand the impact of socioeconomic factors, we did, however, make adjustments for maternal education and mother's gross income. Recommendations for future research include investigating the impact of a wider range of paternal factors in a larger sample, perhaps by linking data from all Nordic countries [37]. A larger sample would further allow for investigating possible interactions between paternal and maternal origin.

A foreign-born father was associated with increased odds of very preterm birth, SGA, stillbirth, and low Apgar score at 5 minutes in births to migrant women. Our results related to very preterm birth support previous findings from 1 Swedish [9] and 1 Canadian [10] study, the increased odds of SGA is consistent with findings from 1 Swedish study [3], and the increased odds of stillbirth is consistent with findings from 1 Canadian [8] and 1 Australian study [11]. In addition, 1 Swedish study reports an increased risk of low Apgar score in migrant women, irrespective of paternal origin [3]. The slightly lower odds found for moderately preterm birth associated with a foreign-born father compared with a Norwegian-born father is consistent with findings from 1 Canadian study [10] and 1 Swedish study [9]. Both studies report that such findings do not apply to all migrant women, highlighting the

importance of being careful with generalization of migrant women's health and individual needs in maternity care. Being able to identify women at risk of adverse neonatal outcomes and then attend to such risks is clinically very important. Very preterm birth is associated with high-cost neonatal intensive care and long-term complex health needs affecting the infant and the whole family [38]; SGA may reflect a range of maternal or infant health problems in need of medical attention [4,39]; stillbirth is associated with high emotional and economic costs affecting the family, communities, and society [40]; and low Apgar score is associated with infant mortality and morbidity [41].

A partner from the host population might ease the integration process by providing the woman with increased wealth and social capital less likely to be available to couples in which both parents are foreign-born [3,11]. He may further facilitate communication between the woman and caregivers and better guide her through the healthcare system [6,11,42]. It is also possible that a migrant woman may be less prone to experience discrimination or disrespectful care from caregivers [43] if the father is from the host population. Such experiences in the receiving country may be important in maintaining "the healthy migrant effect," i.e., that many migrants are of good health, sometimes even better health than the host population [33]. Alternatively, the positive impact of a Norwegian-born father may be explained by maternal factors, rather than paternal ones. In our study, only a few women from the GBD region North Africa and the Middle East reported a Norwegian-born father of the child, whereas a partner from the host population was more common among women from the following regions: high-income countries and Southeast Asia, East Asia, and Oceania. Women from high-income countries are less likely to experience adverse neonatal outcomes compared with more vulnerable groups of migrant women, such as refugees [44]. However, the protective influence of a Norwegian-born partner may not apply to all migrant women. Some migrant women who marry men from the host population have been found to be vulnerable to exploitation, social isolation, and spousal violence [45], factors that may influence a pregnancy outcome negatively [7,46,47].

In births to Norwegian-born women, a foreign-born father was not associated with increased odds of adverse outcomes other than SGA when compared with a Norwegian-born father. These results are difficult to interpret because SGA does not differentiate between growth-restricted babies and healthy babies who are small because of genetic factors [4,39,48]. Thus, biological variation may explain the increased odds of SGA associated with a migrant parent rather than maternal or infant health problems (e.g., malformations or placental insufficiency) [4,39,48]. Further, maternal factors may explain why we found no increased odds of adverse outcomes other than SGA in births to Norwegian-born women. Unfortunately, the present study did not allow for analyses stratified by maternal origin because of the limited number of adverse cases in each subgroup of migrant women. It is worth noting that when Norwegian-born women's babies had foreign-born fathers, these fathers were more likely to come from other high-income countries with fewer cultural and language barriers [49], particularly another Scandinavian country or an English-speaking country, compared with fathers of babies with migrant mothers.

Our findings related to unregistered paternal origin are in line with previous research from the US, Canada, and Australia, where missing paternal demographics have been associated with adverse outcomes such as preterm birth, low birthweight, low Apgar score, stillbirth, and neonatal death [12–14]. However, none of the studies were specific to migrant women [12–14], and the US study only included twin births [12]. Thus, the current study adds important information relevant when planning care for migrant women in particular. Paternal origin was unregistered in as many as 1 in 5 births to women originating from sub-Saharan Africa. It is possible that a lack of information about fathers of the babies of sub-Saharan African

women and their poorer outcomes might be explained by more limited health literacy [50], distrust in the healthcare system [51,52], or a delay in seeking antenatal care [52–54]. This might also be the case for other migrant women, such as those who have fled from wars and conflicts [44]. For migrants in general, data registration is likely to be hampered by language and communication difficulties [55]. Notably, other studies have excluded individuals for whom paternal country of birth [8–10], race, or ethnicity were missing [56], which seems problematic given our results. Our findings suggest that identifying women for whom paternal demographics are missing may help identify high-risk women in maternity care and should therefore alert clinicians.

Unknown paternal identity was particularly strongly associated with very preterm birth and stillbirth. Women may withhold information on a child's biological father for a variety of reasons, including lack of knowledge about the father, economic reasons, or intentions to protect the child, themselves, or others from potential shame or threats [57]. However, these results should be interpreted with caution because the variable paternal identity obtained from the MBRN has not been investigated previously in relation to adverse neonatal outcomes in migrant women. We were unable to differentiate between births in which paternal identity was unknown to the mother or intentionally not reported by the mother or caregiver or if such information was missing because of poor registration routines. We cannot exclude the possibility that the associations between paternal identity and adverse neonatal outcomes are biased for reasons such as systematic under-reporting for women with adverse neonatal outcomes, particularly when a baby is born premature or stillborn. Given the severity of the findings, further studies on unknown paternal identity in the register are warranted.

In conclusion, we found that a foreign-born father was associated with adverse neonatal outcomes among births to migrant women, but to a lesser degree among births to nonmigrant women, when compared with a Norwegian-born father. Unregistered paternal origin was associated with higher odds of adverse neonatal outcomes in births to both migrant and nonmigrant women when compared with Norwegian-born fathers. Increased attention to paternal origin may help identify women in maternity care at risk for adverse neonatal outcomes.

## Supporting information

**S1 STROBE Checklist. STROBE, Strengthening the Reporting of Observational Studies in Epidemiology**
(DOCX)

**S1 Table. Associations between paternal identity and adverse neonatal outcomes in births to migrant and Norwegian-born women in Norway (1990–2016).**
(DOCX)

## Acknowledgments

We would like to thank the MBRN and Statistics Norway for providing data for this study.

## Author Contributions

**Conceptualization:** Eline S. Vik, Vigdis Aasheim, Roy M. Nilsen, Rhonda Small, Dag Moster, Erica Schytt.

**Data curation:** Eline S. Vik, Vigdis Aasheim, Roy M. Nilsen, Erica Schytt.

**Formal analysis:** Eline S. Vik, Roy M. Nilsen, Erica Schytt.

**Funding acquisition:** Eline S. Vik, Vigdis Aasheim, Erica Schytt.

**Investigation:** Eline S. Vik, Roy M. Nilsen, Erica Schytt.

**Methodology:** Eline S. Vik, Vigdis Aasheim, Roy M. Nilsen, Erica Schytt.

**Project administration:** Eline S. Vik, Vigdis Aasheim, Erica Schytt.

**Supervision:** Vigdis Aasheim, Roy M. Nilsen, Rhonda Small, Dag Moster, Erica Schytt.

**Visualization:** Eline S. Vik, Erica Schytt.

**Writing – original draft:** Eline S. Vik, Roy M. Nilsen, Erica Schytt.

**Writing – review & editing:** Eline S. Vik, Vigdis Aasheim, Roy M. Nilsen, Rhonda Small, Dag Moster, Erica Schytt.

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
