## [Editor Report · Decision Letter 0]

11 May 2020

Dear Dr Vik, 

Thank you for submitting your manuscript entitled "Associations between paternal origin and adverse neonatal outcomes in births to migrant women: a Norwegian population-based study" for consideration by PLOS Medicine.

Your manuscript has now been evaluated by the PLOS Medicine editorial staff as well as by an academic editor with relevant expertise and I am writing to let you know that we would like to send your submission out for external peer review.

Kind regards,

Artur Arikainen,

Associate Editor

PLOS Medicine

---

## [Decision Letter · Decision Letter 1]

11 Jun 2020

Dear Dr. Vik,

Thank you very much for submitting your manuscript "Associations between paternal origin and adverse neonatal outcomes in births to migrant women: a Norwegian population-based study" (PMEDICINE-D-20-01868R1) for consideration at PLOS Medicine. 

[LINK]

In light of these reviews, I am afraid that we will not be able to accept the manuscript for publication in the journal in its current form, but we would like to consider a revised version that addresses the reviewers' and editors' comments. Obviously we cannot make any decision about publication until we have seen the revised manuscript and your response, and we plan to seek re-review by one or more of the reviewers. 

We expect to receive your revised manuscript by Jul 02 2020 11:59PM. Please email us (plosmedicine@plos.org) if you have any questions or concerns.

We look forward to receiving your revised manuscript. 

Sincerely,

Artur Arikainen, 

Associate Editor 

PLOS Medicine

plosmedicine.org

1. Please address all of the reviewers’ comments below.

2. Abstract:

a. Please combine the Methods and Findings sections into one section, “Methods and findings”.

b. Please include basic demographic information for the participants (age etc.).

c. In the last sentence of the Abstract Methods and Findings section, please describe the main limitation(s) of the study's methodology.

d. Please quantify the main results (with 95% CIs and p values).

3. The Data Availability Statement (DAS) requires revision. For each data source used in your study: 

a. If the data are freely or publicly available, note this and state the location of the data: within the paper, in Supporting Information files, or in a public repository (include the DOI or accession number).

b. If the data are owned by a third party but freely available upon request, please note this and state the owner of the data set and contact information for data requests (web or email address). Note that a study author cannot be the contact person for the data.

c. If the data are not freely available, please describe briefly the ethical, legal, or contractual restriction that prevents you from sharing it. Please also include an appropriate contact (web or email address) for inquiries (again, this cannot be a study author).

5. Please use the "Vancouver" style for reference formatting, and see our website for other reference guidelines https://journals.plos.org/plosmedicine/s/submission-guidelines#loc-references. Citations should be in square brackets, and preceding punctuation.

6. Methods:

a. Please give exact date ranges for patient recruitment.

b. Please confirm whether the patient data were fully anonymised prior to access by the authors, or whether patients provided written informed consent for the use of their data in research.

7. Results: 

a. Please quantify all results with 95% CIs and p values.

b. Table 1: Please define ‘GBD’ in the legend.

c. Tables 2 and 3: Please define ‘gwks’ in the legend.

8. Discussion: Please include a clear paragraph (or more) on the strengths and limitations of the study.

9. Line 236: Please replace “significant” with “notable” or similar.

10. Please provide access information (URL, DOI, and/or volume/issue/page) for all references, eg. no 2, 22, 41.

11. Please ensure that the study is reported according to the STROBE guidelines, and include the completed STROBE checklist as Supporting Information. 1 Please add the following statement, or similar, to the Methods: "This study is reported as per the Strengthening the Reporting of Observational Studies in Epidemiology (STROBE) guideline (S1 Checklist)."

The STROBE guideline can be found here: http://www.equator-network.org/reporting-guidelines/strobe/ When completing the checklist, please use section and paragraph numbers, rather than page numbers.

12. Did your study have a prospective protocol or analysis plan? Please state this (either way) early in the Methods section.

a. If a prospective analysis plan (from your funding proposal, IRB or other ethics committee submission, study protocol, or other planning document written before analyzing the data) was used in designing the study, please include the relevant prospectively written document with your revised manuscript as a Supporting Information file to be published alongside your study, and cite it in the Methods section. A legend for this file should be included at the end of your manuscript. 

b. If no such document exists, please make sure that the Methods section transparently describes when analyses were planned, and when/why any data-driven changes to analyses took place. 

c. In either case, changes in the analysis-- including those made in response to peer review comments-- should be identified as such in the Methods section of the paper, with rationale.

---------

Comments from the reviewers:

Reviewer #1: I confine my remarks to statistical aspects of this paper. Unfortunately, I think there are some fairly serious issues to resolve.

First, the authors should not use causal language (e.g. line 49 "influence" but other places too). This is an observational study, causation cannot be inferred. 

Second, the dependent variable for gestation should be gestation time in days or weeks and a spline can be used to evaluate nonlinearity. But, if only the categorized data is available, then ordinal logistic regression should be used. Similarly, Apgar score should not be categorized. Doing so makes an Apgar of 1 the same as one of 6 and a score of 7 the same as 10. That makes no sense.

Third, independent variables should also not be categorized. In *Regression Modelling Strategies* Frank Harrell lists 11 problems with this and sums up "nothing could be more disastrous". Leave age and length of residence in years. 

Fourth, for maternal and paternal origin, the reference group should be Norwegian. First, this is both the largest group and the "norm". Second, using foreign born as the reference puts the other two levels below and above the reference. It is better to have the lowest (or highest) category as the reference.

Peter Flom

Reviewer #2: The manuscript entitled "Associations between paternal origin and adverse neonatal outcomes in births to migrant women: a Norwegian population-based study" is an interesting study to provide insight into the effect of paternal origin and the risk of adverse pregnancy outcomes among migrant woman. However, despite the interesting results, I have some comments on the methodology and the discussion that need to be clarified. 

General comments

The association between missing information and adverse health outcomes (and pregnancy outcomes in particular) has been described previously in many studies. The authors should include a wider description and discussion of the unknown paternal identity and unregistered paternal origin to ensure that readers understand exactly what the researchers studied. The difference between unknown paternal identity, unregistered paternal origin and single mother need to be discussed. That is one of the main focus of the paper, but the authors should clarify the social, health and clinical aspects behind those variables: single mother? not access to social and health rights? …. The unreported father is a proxy for more noteworthy factors.

Some methodological details should be expanded and clarified.

- I was wondering if the missing data were MNAR (Missing not at random) . Did you check if there were associated to adverse outcome? We know that missing data for sociodemographic data are more important among stillbirths. If the data are MNAR, the multiple imputation is more complex, and some assumptions need to be verified (check with a statistician) 

- For the logistic regression, did you check the Goodness of fit of the models and interaction tests?

- The stratification primiparous/multiparous need to be discussed because the profiles are similar. The interaction test could justify the choice of stratification rather than adjustment for parity. 

- Maternal factors as single mother and maternal length of residence are mentioned in the methodology but not analyzed? Why ?

Weakness of the discussion

The authors should include more information in the discussion part to clarify some points to avoid confusion. 

- This study requires a comparison with the non-migrant population. The unknown paternal identity and unregistered paternal origin is probably also associated to increased OR for adverse outcome among non-migrant population. The results do not fully support a conclusion specific to migrant population.

- P16 line 267: the discussion for sub-Saharan Africa women can be generalized for other migrant group (literacy, distrust healthcare system, delay antenatal care,..) because it is not specific to sub-saharan Africa mother. Other studies have shown the high proportion of single mother among sub-Saharan African mother. 

The manuscript is certainly well written and attractive, but the results do not provide a substantial advance over existing knowledge for Plos Medicine. The interpretation of the results needs a wider discussion for their implication to clinicians and policy makers. 

Reviewer #3: This manuscript covers the important issue of paternal origin and migrant primip and multip mothers in conjunction with prematurity, low Apgar score, and stillbirth in Norway. The results when controlling for maternal factors of age, education, and income are not surprising, mothers with foreign-born fathers had a higher risk of very preterm and stillbirth compared with Norwegian-born fathers (regardless of multip or primip mothers). Paternal foreign-born status was also associated with lower Apgar score compared to Norwegian-born fathers. Additionally, unknown paternal origin was linked to higher risk of all adverse birth outcomes for both primip and multip mothers. Finally, also expected, unknown father identity was associated with adverse birth outcomes compared with identified father, regardless of primip or multip mother.

The dearth of research on the impact of fathers on birth outcomes makes this a very important contribution to the literature. Furthermore, given the disparity in prenatal care given as noted in the Methods section, hopefully this further highlights the existing inequity and empowers policy makers and providers to move toward solutions in Norway and beyond.

 There are several major issues that do need to be addressed including why the models did not adjust for any paternal factors, why small for gestational age was not included as an outcome, and why the authors excluded the Norwegian born mothers rather than using them in a comparison with migrant mothers. 

Major issues (although the authors may not be able to do all the analyses mentioned, it would be helpful to 1. Add analyses, 2. Add to a limitation, or 3. Note this is a future direction of the research team):

1. The authors note they do not have access to paternal education which does have an independent effect on birth outcomes. Yet they have paternal age and do not adjust for this factor also known to be associated with birth outcomes, especially stillbirth.

2. Further unlike advanced maternal age which is typically defined as 35 years, advanced paternal age is typically defined at 40. It may be beneficial to add this as a category and further examine this in conjunction with migrant mothers and paternal foreign-born status. Also why no assessment of teen aged parents either mother or father (less than 20?), as these lower extremes of ages have also been associated with poor birth outcomes.

3. It is unclear why small for gestational age (a birth outcome previously having been associated with fathers and stress) was not assessed. Could this be added?

4. Another interesting outcome that should be considered if available is NICU admission. NICU admission is more impactful than low apgar score which is not clearly associated with nor predictive of poor neonatal outcomes.

5. It was also unclear why the authors excluded the Norwegian born mothers. Couldn't this group be assessed for paternal foreign born or not and the outcomes could be instructive in relation to migrant mothers and paternal foreign born status. Additionally, outcomes for the general population as a benchmark would also be important to know for generalizability.

6. Additionally, why was both income and education included in the models, often these two variables are co-linear? Was this assessed? Are they both necessary in the models?

7. There is no discussion of race and ethnicity. In the US, there is the "healthy immigrant" effect, possible as certain races/ethnicities have not had the weathering from endemic racism. Could a migrant mother and foreign born father have additional discrimination from their race/ethnicity aside from country of origin? Was there data on this? Also consider referencing Collins work (one reference is Collins JW Jr, Wu SY, David RJ. Differing intergenerational birth weights among the descendants of US-born and foreign-born Whites and African Americans in Illinois. Am J Epidemiol. 2002; 155:210-6.)

8. How long were the mothers and father in Norway for and why was this not adjusted for as the authors mention they calculate maternal length of residence and categorize it, but I did not see that data nor why this was not in the models.

Other Minor Edits/By Section:

Abstract

1. Results - Line 46-48 "Unregistered paternal origin and unknown paternal identity were both associated with increased odds of adverse neonatal outcomes." Please clarify in comparison with?

Introduction

1. The introduction seems to simplify a very complex body of literature where paternal identification and origin is intertwined with marital status, race/ethnicity, social economic status and support. This study approaches this by separating out countries of origin of the mother. However, there should be some acknowledgement of additional known paternal effects and birth outcomes.

2. I believe the introduction would be strengthened by more explicitly delineating the gaps in literature, how this study will address the gaps in a novel way in the Introduction.

Methods:

1. Could the choices of what did and did not go into the models be more detailed? Why not consanguinity or paternal age?

2. Can you explain the exclusion factor of second-generation migrant women, does this mean the women were excluded if they were born in Norway but their parents were not or does it mean that the migrant woman was foreign born but her parents were from Norway? What does women with mixed background mean? One parent was from Norway and one not?

3. The information provided in the Setting Section of the Methods was very helpful.

4. It might be helpful to provide examples of which high income countries are most common locations for migrants to originate from.

5. Is there data on previous preterm birth-a big predictor of preterm delivery in the future among the multi-parous, if not should be listed in Discussion as limitation.

6. Any data on maternal high risk pregnancy? Pre-existing morbidities OR pregnancy related diseases-preclampsia or diabetes or obesity that could increase risk of stillbirth and rate of prematurity? If not, should be a limitation in Discussion

7. What about substance use, especially tobacco use which may be higher or lower in certain countries mothers and fathers migrate from? If not, should be in Discussion.

8. Why not report in addition to very and moderately preterm rates, the overall preterm rate?

Results:

1. No bivariate analyses of frequencies reported, only OR and aOR. However, would be helpful to know if the frequencies differ among groups. Were the data in Table1, Figure 2 compared/analyzed? If so should be denoted

2. Why do authors think slightly increased risk of Norwegian-born fathers before adjustment of having moderate preterm birth? 

3. For Table 3 Does known father mean known and could be either foreign born or norwegian? If so these might be separated? Either way should be clarified.

Discussion:

1. I think the Discussion could benefit to some restructuring such that limitations are noted together in serial paragraphs where limitations are not just listed but as to why this study is still valid despite the limitations. Foloowed by a strengths paragraph explaining why this study is well designed, novel and important contribution. The current strength paragraph is mixed with limitations. I have noted above some possible limitations to explore. 

2. Line 264 which notes that the US study only examined twins. There are other US studies not only of twins. One example is Ma S. Paternal Race/Ethnicity and Birth Outcomes. December 2008, Vol 98, No. 12 | American Journal of Public Health. 

3. It may be helpful to discuss more of the biological underpinings of the findings-toxic stress and related inflamation which can lead to poor birth outcomes, how support from a father regardless of marriage may mitigate the inflammation etc. 

4. Do the authors surmise that a father from the host population makes a difference due to anything specific to infrastructure in Norway that might not exist in the countries that published the studies the authors cite as background? If so this might also help the reader know why this study is different/may be novel?

[LINK]

---

## [Decision Letter · Decision Letter 2]

22 Jul 2020

Dear Dr. Vik,

Thank you very much for submitting your manuscript "Associations between paternal origin and adverse neonatal outcomes in births to migrant women: a Norwegian population-based study" (PMEDICINE-D-20-01868R2) for consideration at PLOS Medicine. 

Your paper was evaluated by a senior editor and discussed among all the editors here. It was also discussed with an academic editor with relevant expertise, and re-sent to independent reviewers. The reviews are appended at the bottom of this email and any accompanying reviewer attachments can be seen via the link below:

[LINK]

In light of these reviews, I am afraid that we will still not yet be able to accept the manuscript for publication in the journal in its current form, but we would like to consider a revised version that addresses the reviewers' and editors' comments. Obviously we cannot make any decision about publication until we have seen the revised manuscript and your response, and we plan to seek re-review by one or more of the reviewers. 

We expect to receive your revised manuscript by Aug 12 2020 11:59PM. Please email us (plosmedicine@plos.org) if you have any questions or concerns.

We look forward to receiving your revised manuscript. 

Sincerely,

Artur Arikainen, 

Associate Editor 

PLOS Medicine

plosmedicine.org

1. Please address reviewer #3’s comments below.

2. Title: Please update to: “Associations between paternal origin and adverse neonatal outcomes in births to migrant women in Norway: a population-based cohort study”

3. Short Title: Please update to: “Paternal origin and neonatal outcomes in births to migrant women in Norway”

4. Financial Disclosure: Please confirm whether all funders had no role in study design, data collection and analysis, decision to publish, or preparation of the manuscript.

5. Data Availability Statement (DAS): Please also include an appropriate contact (web and/or email address) for data access inquiries (this cannot be a study author).

6. Abstract: 

a. In lines 58-60, please mention the word “limitations” specifically.

b. For p values greater than 0.001, please give the exact value.

c. Lines 50-52, please quantify these results with 95% CIs and p values, even if not significant.

d. Line 51: Please avoid use of italics for emphasis.

e. Line 61: Please begin with “We found that…”

f. Conclusion: Please add a brief note on the ramifications of your study.

7. Author Summary:

a. Lines 80 and 82: Please update to: “…father of the child…”

b. Line 82: Please avoid use of italics for emphasis.

8. Please remove spaces from within the brackets of your citation callouts, eg: “…such as low birth weight [3,4]…”

9. Methods: Please mention explicitly that: “There was no prospective protocol or analysis plan.”

10. Please include the completed STROBE checklist as Supporting Information.

---------------

Comments from the reviewers:

Reviewer #1: The authors have addressed my concerns and I now recommend publication

Peter Flom

Reviewer #3: The authors have done a significant amount of work to respond well to my critiques, especially adding the Norwegian mothers. I still have one major concern remaining:

1. I strongly urge the authors to reconsider their lack of analysis of size for gestational age. Similar to their argument for the concerns of birthweight for gestational age, prematurity is multi-factorial (maternal social factors, maternal genetic factors, maternal chronic disease/overall health, fetal health). Regardless, in general, it is accepted that small for gestational age is a poor birth outcome as prematurity is, and both have been linked to not only maternal factors but also paternal factors. In fact, in the Introduction section, the authors refer to low birth weight (less informative that size for gestational age) as an adverse birth outcome. There are very few birth outcomes based on vital statistics to assess, and size for gestational age or even term low birth weight are both meaningful. The authors have an opportunity to study birthweight for gestational age in this unique population. In fact, just recently, these same authors using a migrant mother/Norwegian birth cohort and birth outcomes similar to this manuscript (but without paternal information) also examined SGA (small for gestational age) status (Vik et al. Country of first birth and neonatal outcomes in migrant and Norwegian-born parous women in Norway: a population based study. BMC Health Services Research (2020) 20:540). It would be interesting for the authors to refer to their BMC Health Services Research publication and compare/contrast SGA and other birth outcome in the current Results and Discussion sections as well.

[LINK]

---

## [Decision Letter · Decision Letter 3]

25 Aug 2020

Dear Dr. Vik,

Thank you very much for re-submitting your manuscript "Associations between paternal origin and adverse neonatal outcomes in births to migrant women in Norway: a population-based cohort study" (PMEDICINE-D-20-01868R3) for review by PLOS Medicine.

I have discussed the paper with my colleagues and the academic editor and it was also seen again by one reviewer. I am pleased to say that provided the remaining editorial and production issues are dealt with we are planning to accept the paper for publication in the journal.

[LINK]

We look forward to receiving the revised manuscript by Sep 01 2020 11:59PM. 

Sincerely,

Artur Arikainen, 

Associate Editor 

PLOS Medicine

plosmedicine.org

Requests from Editors:

1. Title: Please update to: “Paternal country of origin and adverse neonatal outcomes in births to foreign-born women in Norway: a population-based cohort study”

2. Data Availability and Funding Statements: Please update these fields in the online submission form to include the previously requested information. You can remove the sections on lines 17-27.

3. Abstract:

a. Please quantify these results with 95% CIs and p values: “Findings related to unregistered paternal origin were similar for Norwegian-born women.”

b. Please add another limitation at line 66, eg. inability to account for all lifestyle factors.

c. Please correct the data for these statements:

i. “In migrant women, unregistered paternal origin was associated with increased odds of … low Apgar score (3.4% vs 1.5% aOR 1.23; CI 1.99-1.50, p<0.001)” – aOR is given as 2.23 in Table 2.

ii. “In births to Norwegian-born women, a foreign-born father was associated with … decreased odds of moderately preterm birth (4.3% vs 4.4%, aOR 0.95; CI 0.93-0.99, p=0.015)” – CI given as 0.91-0.99 in Table 3.

d. (Please double check all data in the manuscript for accuracy.)

4. Line 100: Please clarify if you mean “international migrant”.

5. Tables:

a. Please ensure all abbreviations are defined in the footnotes.

b. We recommend replacing the vertical bar symbol (|) for footnotes with another symbol (or superscript letters) for clarity, where used.

6. Please ensure that all references are complete – 18 and 49 appear to be missing journal names.

Comments from Reviewers:

Reviewer #3: The authors have sufficiently responded to my critiques.

[LINK]

---

## [Editor Report · Decision Letter 4]

18 Sep 2020

Dear Mrs. Vik, 

On behalf of my colleagues and the academic editor, Dr. Jenny E Myers, I am delighted to inform you that your manuscript entitled "Paternal country of origin and adverse neonatal outcomes in births to foreign-born women in Norway: a population-based cohort study" (PMEDICINE-D-20-01868R4) has been accepted for publication in PLOS Medicine. 

PRODUCTION PROCESS

PRESS

PROFILE INFORMATION

Thank you again for submitting the manuscript to PLOS Medicine. We look forward to publishing it. 

Best wishes, 

Artur Arikainen, 

Associate Editor 

PLOS Medicine

plosmedicine.org